# Obtaining Hyperspectral Signatures for Seafloor Massive Sulphide Exploration

**Øystein Sture** [1,*] **, Ben Snook** [2] **and Martin Ludvigsen** [1,3,4]

1   Department of Marine Technology, Faculty of Engineering, Norwegian University of Science and Technology (NTNU), Otto Nielsens Veg 10, 7052 Trondheim, Norway; martin.ludvigsen@ntnu.no
2   Department of Geoscience and Petroleum, Faculty of Engineering, Norwegian University of Science and Technology (NTNU), Sem Sælands Veg 1, 7491 Trondheim, Norway; ben.snook@ntnu.no
3   Centre for Autonomous Marine Operations and Systems, Department of Marine Technology, Norwegian University of Science and Technology (NTNU), Otto Nielsens Vei 10, NO-7491 Trondheim, Norway
4   Arctic Technology Department, University Centre in Svalbard (UNIS), P.O Box 156, NO-9171 Longyearbyen, Norway
*   Correspondence: oystein.sture@ntnu.no

**Abstract:** Seafloor massive sulphide (SMS) deposits are hosts to a wide range of economic minerals, and may become an important resource in the future. The exploitation of these resources is associated with considerable expenses, and a return on investment may depend on the availability of multiple deposits. Therefore, efficient exploration methodologies for base metal deposits are important for future deep sea mining endeavours. Underwater hyperspectral imaging (UHI) has been demonstrated to be able to differentiate between different types of materials on the seafloor. The identification of possible end-members from field data requires prior information in the form of representative signatures for distinct materials. This work presents hyperspectral imaging applied to a selection of materials from the Loki's Castle active hydrothermal vent site in a laboratory setting. A methodology for compensating for systematic effects and producing the reflectance spectra is detailed, and applied to recover the spectral signatures from the samples. The materials investigated were found to be distinguishable using unsupervised dimensionality reduction methods, and may be used as a reference for future field application.

**Keywords:** seafloor sulphides; underwater hyperspectral imaging; spectroscopy; reflectance; Norway; Arctic Mid-Ocean Ridge

## 1. Introduction

Remote sensing with hyperspectral and multi-spectral technologies has seen wide use in prospecting for minerals and hydrocarbons on land [1]. Materials altered through hydrothermal processes may be associated with mineral deposits, and have been a focus of classical multi-spectral imaging such as the Landsat Thematic Mapper (TM). In particular, alunite and hydrothermal clay have distinct reflective and absorptive properties at the wavelengths centred at 1.65 μm (TM5) and 2.215 μm (TM7). Iron oxides and sulphates have low blue reflectance (TM1) and high red reflectance (TM3) in the visible spectrum. The ratios between these pairs have been used to discriminate in prospecting for terrestrial deposits of hydrothermal origin [2].

Electromagnetic waves with wavelengths in the near-infrared (NIR) and above are difficult to utilise underwater due to rapid attenuation or absorption by the water medium. However, the discrimination of different materials based on reflective properties using visible/near-infrared imaging spectroscopy (VNIR; 0.4 μm to 1.0 μm) may still be possible [3]. Determining possible

end-members from spectral responses requires prior knowledge of the materials' optical properties (i.e., their reflectivity). These signatures can be obtained in a controlled environment, where the spectra of the lamps, wavelength-dependent attenuation in water, and scattering of light can be accounted for [4].

### 1.1. Seafloor Massive Sulphides

Seafloor massive sulphides (SMS) is a style of mineralisation currently ongoing at active hydrothermal vents (black and white smokers). The deposits are formed in locations where the presence of faults and cracks in the crust allows water to seep into the seabed and ultimately promote hydrothermal circulation as the water draws closer to a heat source and is warmed. Depending on the type of surrounding host rock, the seawater can become acidic, anoxic, and/or alkali-rich. These fluids are capable of leaching metals (e.g., Cu, Fe, Mn, and Zn) from the surrounding rocks, which are then carried to the surface [5,6]. While a substantial amount of these minerals might be carried away with the currents, a fraction is deposited directly where the fluids emerge or in the near-surface stockwork, as the fluids are exposed to cold sea water. Over time, these locations can accumulate significant deposits rich in valuable minerals such as zinc, copper, silver, and gold [7]. Such deposits are known to exist at depths of 800–5000 m [8]. The deposits are located near the seafloor, and some have been shown to contain similar metal contents to high-grade land-based deposits [9]. Seafloor massive sulphide (SMS) deposits are estimated to contain significant amounts of metal contents [8], and may therefore become important resources in the future.

The European Commission considers the primary raw materials found in SMS deposits to be of high economic importance whilst having a high reliance on imports [10,11]. Demand for copper is driven by population and per-capita income; the effect of a decrease in population growth may be offset by the increased per-capita income of populous countries such as China and India [12]. The availability of these raw materials is essential for the production of goods and the further development of eco-efficient technologies. The demand for electrical motors in industrial applications and electrical vehicles will for example lead to the additional consumption of copper [13].

### 1.2. Related Work

Underwater hyperspectral imagers have seen application in marine biology, for instance to detect the presence of toxic compounds in corals [14], and identifying groups of algae [15]. Recently, hyperspectral imagery has also been used in archaeology for detection of man-made objects of interest [16]. Spectrometers and hyperspectral imagers have been deployed on a wide range of underwater platforms including landers, remotely operated vehicles (ROVs) and autonomous underwater vehicles (AUVs) [17–20].

Field trials of underwater hyperspectral imaging (UHI) on marine minerals have previously been performed on SMS at the Trans-Atlantic Geotraverse (TAG) hydrothermal vent field, and on manganese nodules in the Peru Basin [17,21]. These trials demonstrated the ability to discriminate between different groups of material. The materials were compared using pseudo-reflectances—spectra which were normalised to the median spectral response within each transect to compensate for the source light spectrum and attenuation. For this reason, the signatures do not generalise to areas where the median spectral response of the seabed differs. In this work, we aim to remedy this issue by outlining an experimental setup to compute the absolute reflectances of target materials, thus allowing for the direct comparison of measurements made at distinct sites or laboratory measurements.

### 1.3. Underwater Light Propagation

In this section, we briefly introduce a model of underwater light propagation to support discussion around the key simplifications and assumptions made in this work. We describe the propagation of light in terms of geometrical optics, ignoring effects such as diffraction, wave interference and secondary emission of photons from the target material. The light propagation is described in terms

of rays, which are reflected, refracted and absorbed depending on the incidence angle, medium and reflective properties of the material. The ultimate goal is the removal of effects not pertinent to the reflective properties of the target material from the irradiance, as measured by the sensor. In the following, $\langle \cdot, \cdot \rangle$ is taken to be the dot or scalar product, and $\| \cdot \|$ is the Euclidean norm. The defined vectors and areas are illustrated in Figure 3.

We consider light emitted from a single source for a given wavelength $\lambda$, reflected onto the image plane of the camera via a differential area $dA$ in the object plane. We first consider the irradiance incident on the differential area from the radiant intensity originating from a point source, $E_{dA}$ W/m$^2$. The source strength may vary depending on the source direction represented in polar coordinates, $\theta_s$, $\phi_s$, relative to $\vec{n}_s$:

$$E_{dA} = \mathcal{I}_s(\lambda, \theta_s, \phi_s) d\omega_s \frac{\langle \hat{n}, \hat{r}_s \rangle}{\|\vec{r}_s\|^2} \exp(-c(\lambda)\|\vec{r}_s\|). \tag{1}$$

The unit area, $dA$, receives a flux according to the radiant intensity, $\mathcal{I}_s$ W sr$^{-1}$, for the unit solid angle $d\omega_s$ subtended by the projected surface area. Here, $\langle \hat{n}, \hat{r}_s \rangle$ denotes the dot product between the unit directional vectors, equivalent to the cosine of the angle, which accounts for Lambert's cosine law. The inverse square law accounts for losses associated with spherical propagation from a point source. If the directionality of the light source is determined by physical constraints, which affects all wavelengths equally, one can divide the directivity into a reference source level and beam pattern $B$, such that $\mathcal{I}_s(\lambda, \theta_s, \phi_s) = \mathcal{I}(\lambda)B(\theta_s, \phi_s)$. The light is attenuated exponentially with distance throughout the medium, by processes of absorption and scattering, the rate of which is determined by the attenuation coefficient of the water, $c(\lambda)$.

To describe the relationship between the irradiance incident on the differential surface and the amount of radiance reflected, the bidirectional reflectance distribution function (BRDF) is defined as a function of wavelength, incidence angle and viewing angle (Equation (2)):

$$F_r(\lambda, \theta_{ns}, \theta_{nc}) = \frac{dL(\lambda, \theta_{nc})}{dE_{dA}(\lambda, \theta_{ns})} \left[ \frac{Wsr^{-1}m^{-2}}{Wm^{-2}} \right]. \tag{2}$$

For a Lambertian scatterer, where the incident light is distributed equally across all hemispherical directions, this becomes [22]:

$$F_r(\lambda) = \frac{\rho(\lambda)}{\pi}. \tag{3}$$

Here, $\rho(\lambda)$ is the albedo, or diffuse reflectivity, of the target material. For a perfectly Lambertian scatterer, the reflectance and albedo are equivalent. Now, considering the differential surface element as a source of outgoing irradiance, we can formulate the irradiance incident on the camera sensor:

$$E_c = E_{dA} F_r(\theta_{ns}, \theta_{nc}) \frac{T_L \Omega_L}{m^2} \langle \hat{n}, \hat{r}_c \rangle \exp(-c\|\vec{r}_c\|). \tag{4}$$

Here, $T_L(\theta_c)$ is the transmission loss of the lens in a given camera direction, and $\Omega_L$ is the solid angle subtended by the circular aperture of the camera, as seen from the differential surface element. The magnification, $m$, accounts for the change in surface area when projected onto the image plane. We can approximate $\Omega_L$ by projecting the circular aperture as follows [23]:

$$\Omega_L \approx \pi \left( \frac{D}{2} \right)^2 \frac{\langle \hat{n}_c, -\hat{r}_c \rangle}{\|\vec{r}_c\|^2}. \tag{5}$$

This, inserted into Equation (4), becomes as follows for a single wavelength with the input variables omitted for the sake of brevity

$$E_c \approx \mathcal{I}_s d\omega_s F_r \frac{\pi D^2 T_L}{4m^2} \frac{\langle \hat{n}, \hat{r}_s \rangle}{\|\vec{r}_s\|} \frac{\langle \hat{n}_c, -\hat{r}_c \rangle}{\|\vec{r}_c\|^2} \langle \hat{n}, \hat{r}_c \rangle \exp(-c(\|\vec{r}_s\| + \|\vec{r}_c\|)) \tag{6}$$

By simplifying the coordinate system, replacing the dot products with cosines, neglecting the thickness of the lens, and expressing the magnification in terms of the focal number and focal length, one can arrive at the direct component of the Jaffe–McGlamery computational model [23–25].

The model does not account for light that is scattered but still recorded by the sensor. These effects can be divided into backscattering, where the light returns to the sensor without interacting with the material, and small-angle forward scattering, in which the transmission of light from the object is displaced but still measured. We assume that the medium has sufficiently low turbidity, such that these effects can be neglected. In a laboratory setting, where we have control of the water quality, this assumption can be made.

Note also that the target application is hyperspectral imaging, which necessitates the presence of additional optical components such as prisms, slits or mirrors to split the light into its constituent wavelengths compared to regular cameras. Accurately modelling these effects may require the transmission loss to vary according to the camera viewing angle and/or wavelength, $T_L(\theta_i, \lambda)$.

## 2. Materials and Methods

The following section describes the laboratory equipment used, the experimental setup and configuration, as well as the samples investigated. The imager records light as reflected by the target material, but does not directly observe the target reflectance. The steps taken to recover reflectances from the raw digital counts recorded by the imager are therefore outlined. This is done by relating new measurements to a measurement of a known material at the same height and across-track position. The laboratory experiment was performed at the Trondheim Biological Station (NTNU, Trondheim, Norway).

### 2.1. Samples

The Mid-Atlantic Ridge (MAR) extends from the north of Greenland through Iceland and Jan Mayen all the way to the South Atlantic Ocean, close to Bouvet Island. The Arctic Mid-Ocean Ridge (AMOR) is the northern part of this ridge, where the North American and Eurasian tectonic plates meet. Mohn's Ridge is defined as a slow- to ultraslow-spreading ridge [26]. More than 80% of undiscovered SMS deposits are suggested to be located on these types of ridges [27]. These types of ridges are characterised as having less-frequent vent sites, but with a potential for generating larger deposits than fast-spreading ridges [28].

The samples studied in this work were collected in 2016 at Loki's Castle on the AMOR [29]—a site of known active hydrothermal venting. The vent site is situated at 2400 m depth, on an axial volcanic ridge (AVR) [30]. The vent site consists of two mounds, each approximately 150 m across and 30 m high, capped by five active chimneys releasing fluids at 320 °C. Sample boulders were collected with an ROV from collapsed chimney fragments near the surface. Upon recovery on deck, the samples were vacuum-packed in double nitrogen-flushed plastic and stored in a freezer at −21 °C to mitigate the oxidation of the sulphides. These samples have previously been investigated and were found to contain a diverse mineralogy with highly heterogeneous macro- and micro-textures; the sulphides consist of isocubanite and chalcopyrite intergrowths and sphalerite and abundant pyrite/marcasite, with amorphous silica and barite as gangue [31]. Detailed geochemistry (from ICP-ES/MS) and mineralogy (from XRD) (after [31]) are provided in Appendices A and B, respectively.

The samples used in the experiment are illustrated in Figure 1. The low-grade SMS sample is largely composed of barren gangue, with lenses of relatively richer sulphide-bearing phases. The high-grade SMS material is composed entirely of this rich sulphide phase. The basalt and the mudstone are not mineralised, but were included as they are lithologically relevant to this style of mineralisation.

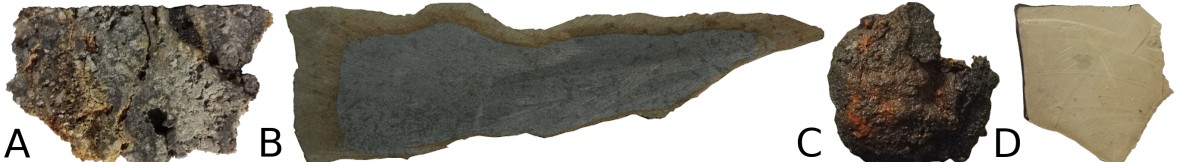

**Figure 1.** The samples used in the experiment, representing variations in mineralised material and in potential host rocks: (**A**) low-grade SMS; (**B**) basalt; (**C**) high-grade submerged massive sulphide (SMS); and (**D**) mudstone. Samples B and C are investigated from multiple sides.

*2.2. Equipment*

The specifications of the hyperspectral imager used in this work are summarised in Table 1. The imager supports operation up to the spatial resolution and radiometric resolution listed, but also allows for binning of adjacent pixels or bands. Binning sacrifices resolution for an increased signal-to-noise ratio (SNR). Many samples reflect only a small fraction of the incident light, and binning may therefore be necessary even in a laboratory setting. In this work, a binning of two was used, both spectrally and spatially. A pair of halogen-based DeepSea Multi-SeaLite lamps with 250-W bulbs from Iwasaki Electric Co. lamps were used to illuminate the samples. Halogen was chosen due to the increasing illumination towards the upper part of the visible spectrum, which helps counteract the higher attenuation in water towards the near-infrared (NIR).

**Table 1.** Ecotone UHI-3 specification.

| | |
|---|---|
| Imager | 12 mm fore objective, spectrograph, sCMOS |
| Size (H × W) | 355 mm × 135 mm (cylindrical) |
| Weight | 11 kg in air, 6 kg in water |
| Depth rating | 3000 m |
| Operating distance | 0.2 to 7 m |
| Operating temp | $-5\,°C$ to $30\,°C$ |
| Power | 12 V to 36 VDC, max. 35 W |
| Frame rate | 1 to 80 Hz |
| Spectral range | 380–800 nm |
| Diff. lim. spec. res. | 5 nm |
| Spatial res. | 1920 pixels |
| Radiometric res. | 12 bits per band |

An Ocean Optics Jaz spectrometer and Ocean Optics HL-2000 light source was used together with a Labsphere Spectralon® diffuse reflectance standard (50%) for calibration.

*2.3. Laboratory Setup*

Properties of materials such as translucency (e.g., quartz/silica) and microscopic surface roughness have been shown to have an effect on the reflective properties of the material when submerged in water. The shape of the angular response is largely unchanged given that the material does not have a significant amount of translucent particles, but may also exhibit an effect known as wetting where the material becomes darker under water [32]. While practical, recording the measurements in-air will therefore not necessarily match the in-water measurements across different materials. For this reason, the samples were imaged with both the samples and camera fully submerged in a water-filled tank. Salt water was taken from an outlet in the laboratory, pumped from 100 m depth from the fjord adjacent to the laboratory and sand-filtered twice to reduce the effect of organic material and other pollutants. The interior of the tank was covered in a black fabric to reduce the residual effects of light via the edges of the white plastic tank. The material was selected based on the following criteria; a dark colour with a matte surface, durable, and non-reactive with water.

The hyperspectral imager and source lights were mounted on a motorised rig, enabling smooth motion between the two ends of the tank. The imager acquired lines with an exposure time of 60 ms

moving at approximately 4.3 cm s$^{-1}$, resulting in a movement of 0.258 cm per acquired line. The lamps were mounted 20 cm on both sides of the UHI, facing straight down. The distance from the bottom of the tank to the camera and light was 58 cm. At one end of the tank, an inclined calibration plate was placed to enable compensation for the attenuation in water and source light geometry at different heights. An illustration of the setup is shown in Figure 2. The experimental setup was similar in design to one that was previously used for the estimation of coral toxicity [14]. One notable difference is the omission of the submerged Spectralon reflectance standard, instead opting to use the plastic backplate of the sample basket for the same purpose. Spectralon calibration plates have previously been found to deviate from a perfect Lambertian scatterer when submerged, and exhibited an increased reflectance at narrow viewing angles compared to a perfectly diffuse scatterer (10%) [33]. The use of submerged Spectralon standards is still valid, but a correction to the factory reference data is needed. This can either be done based on values from the literature, or using a reflectance probe on the submerged plate under the assumption that the attenuation due to the short distance from probe to plate is negligible. In this work, the submerged plate was calibrated relative to an in-air Spectralon plate.

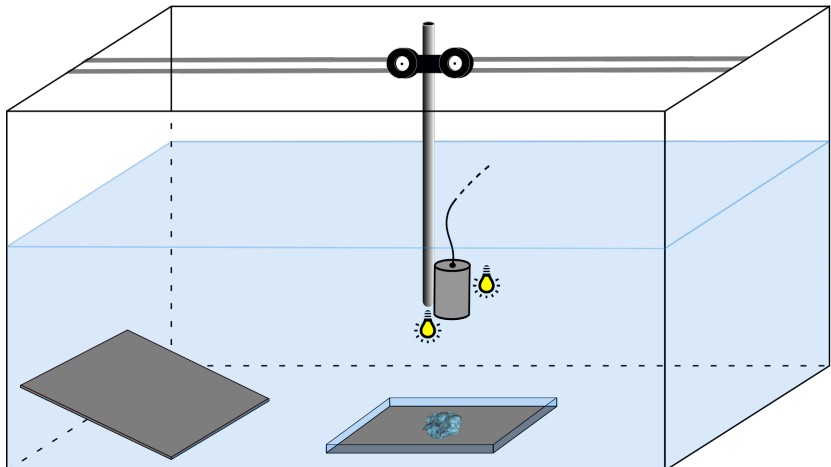

**Figure 2.** Illustration of the laboratory tank setup. The illustration depicts the tank, moving camera rig, hyperspectral camera, halogen lamps, samples in a submersible crate and tilted calibration plate.

The following measurements were made prior to submerging the samples in the tank. Using an in-air spectrometer and diffuse reflectance standard, the reflectance of the flat reference plate to be used in the tank was measured. The measurements of both the diffuse standard and reference plate were performed at the same inclination as in the tank, to account for non-Lambertian reflectances. A measurement was then taken with the lights turned off, such that any persistent effects from external light sources, background thermal radiation and electrical noise could be removed. This measurement was subtracted from all subsequent measurements.

### 2.4. Inclined Reference Plate

The samples under investigation vary in their height above the bottom of the tank. This alters the distance the light has to travel through the medium as well as the direction towards the light sources. To account for variation in nominal light intensity due to attenuation and lamp directionality, an inclined reference plate was imaged as a part of the experiment. This may need to be repeated depending on the duration of the experiment, how reactive the samples were with the water and possible temperature fluctuations. While the previously presented model of light propagation does not take the lens and optics of the camera into account, the main losses during the radiative transfer from light source to camera are described and is used as a reference forward model.

The vectors between the camera, reference plate and light sources can be derived through basic trigonometry for this setup. The use of vector notation allows for changes to the position of the lights

relative to the camera or the surface orientation without altering the equations. All vectors can be expressed in terms of the height and viewing angle, provided that the height of the plate is known. For the following equations, the origin of the coordinate system coincide with the camera coordinate frame (i.e., centred on the camera opening). The $x$-axis is oriented towards the end of the tank with the reference plate, $z$-axis downwards, and $y$-axis according to a right-handed coordinate system. The hyperspectral imager in this work is a line-scanner, where each exposure captures $n_w$ pixels each with $n_d$ bands. Each pixel must therefore have an associated unit vector, $\hat{r}_{c,i}, i = 1 \dots n_w$, describing the measurement direction. Since the field of view of the camera is aligned with the $yz$-plane, the unit vectors can be obtained by a rigid rotation about the $x$-axis. The amount of rotation is given by the exit angles after correcting for refraction through the lens, $\theta_{c,i}$.

$$\hat{r}_{c,i} = \begin{bmatrix} 1 & 0 & 0 \\ 0 & c_{\theta_{c,i}} & -s_{\theta_{c,i}} \\ 0 & s_{\theta_{c,i}} & c_{\theta_{c,i}} \end{bmatrix} \begin{bmatrix} 0 \\ 0 \\ 1 \end{bmatrix} = \begin{bmatrix} 0 \\ -s_{\theta_{c,i}} \\ c_{\theta_{c,i}} \end{bmatrix}, \tag{7}$$

where $c_{\theta_{c,i}}$ and $s_{\theta_{c,i}}$ denote the cosine and sine, respectively. The intersection of this line with a plane described by a normal vector $\hat{n}_p$ and a point on the plane $p_p$ can be calculated by the following equations. Since the origin of the coordinate system is assumed to be at the start of the ray, the intersection point is equal to the camera viewing vector, $\vec{r}_{c,i}$:

$$\vec{r}_{c,i} = -\hat{r}_{c,i} \frac{\langle p_p, \hat{n}_p \rangle}{\langle \hat{r}_{c,i}, \hat{n}_p \rangle}, \tag{8}$$

where the direction is reversed to match Figure 3. The vector from the intersection point to the light sources can now be calculated as:

$$\vec{r}_{s,j} = \vec{r}_{c,i} - p_{s,j}, \tag{9}$$

where $j = 1 \dots n_s$ denotes the index of the light source, for the setup presented here $n_s = 2$. The polar coordinates of the beam-pattern, $\theta_s$ and $\phi_s$, can be computed from this vector. It may be practical to define the beam pattern relative to the main axis of the source light.

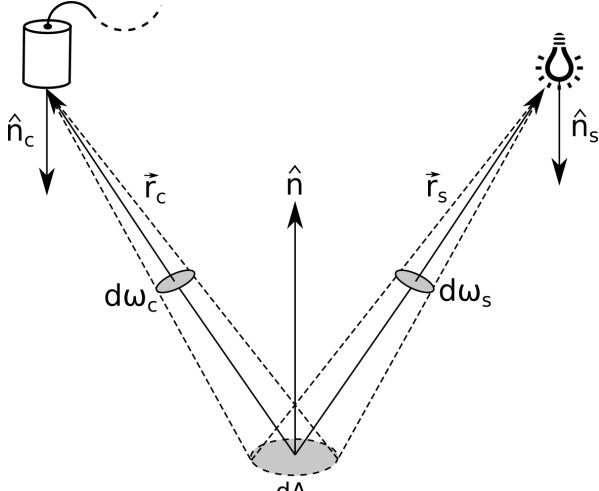

**Figure 3.** Illustration for underwater light propagation and imaging. The unit normal vectors of the camera, surface and light source are used to indicate their orientation. The vectors $\vec{r}_c$ and $\vec{r}_s$ indicate the direction and distance between the surface and camera/source, respectively. The areas $d\omega_c$, $d\omega_s$ and $dA$ denote the solid angle subtended by the camera aperture, the solid angle of the projected surface area and a differential surface area, respectively.

During the experiment, the plate was placed at one end of the tank, inclined at an angle of 33.6 deg. The difference between the lowest and highest points of the plate was 12.65 cm. Using the formulas above, the relevant variables could be calculated based on the position over the plate, by setting $p_p = [0, 0, h]$, where $h$ is the height above the reference plate. For the setup used in this work, the height increased linearly as the camera moved across the plate.

## 2.5. Noise Properties

The model for underwater light propagation presented in Section 1.3 does not include a noise term. Noise in the UHI measurements can originate from a mixture of external sources, such as temperature fluctuations in source intensity and inherent sensor noise. To determine the properties of the noise, the camera rig was placed stationary over the reference plate, and 50 frames were captured across the same line. The mean measurement was subtracted for each across-track pixel $j = 1 \ldots n_w$ per wavelength $k = 1 \ldots n_d$. The noise was verified to be normally distributed with an identical variance as a function of across-track position, but was not found to be isotropic as a function of wavelength. Figure 4 shows the variance as a function of wavelength for the top and bottom of the calibration plate along with the mean intensity at the centre of the measurement line.

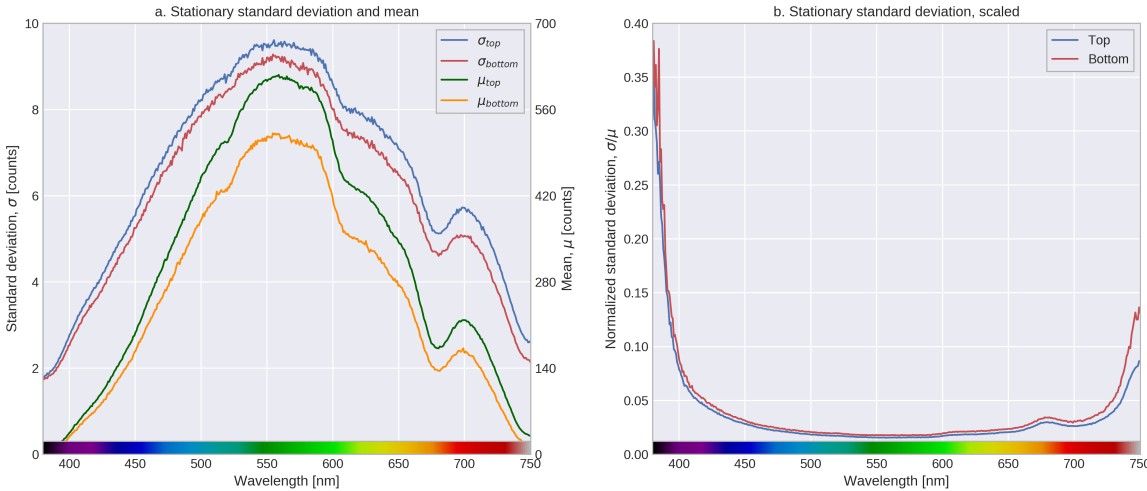

**Figure 4.** (**a**) Standard deviation and mean as a function of wavelength at the top and bottom of the calibration plate. The standard deviations were for the most part in the range of 2–4% of the measured digital counts. The mean measurements were taken across the centre line. The ambient noise level (i.e., with lights turned off) was $\sigma_{\mathrm{amb}} = 1.6$ digital counts across all wavelengths. (**b**) The same standard deviations at the bottom and top of the plate, normalised by the respective mean.

Noise that is not additive, but multiplicative, is affected by the relative strength of the source lamp, attenuation across wavelengths and reflectivity. The difference in variance between the top and bottom of the plate can be explained by different attenuation lengths and source directions.

## 2.6. Non-Parametric Regression

To calculate the reflectance of objects of interest, one must account for changes to the measured irradiance not caused by the reflective properties of the material. By measuring the intensity of light over a material of known reflectance at different heights, we have a reference level to compare new measurements to. In this section, a non-parametric regression method is outlined for the reference measurements, to be able to interpolate and mitigate the effect of measurement noise. Using Equations (8) and (9), all input variables in the model can be computed from the camera viewing angle $\theta_{c,i}$, vertical distance to bottom ($h$) and bottom normal vector $\hat{n}_p$. The latter can be factored out assuming that the bottom inclination is known. The measured irradiance over the tilted

calibration plate can therefore be seen as a function of vertical distance and viewing angle, plus noise. Gaussian process (GP) regression is applied to model the source strength, attenuation, beam pattern and other effects while accounting for the measurement noise of the calibration plate. Whereas a Gaussian probability distribution describes random variables, a GP is a generalisation of this concept to stochastic processes. In this context, the process is a latent function $f(\mathbf{x})$, fully described by a mean function and covariance kernel as functions of the regressor (input variables):

$$f(\mathbf{x}) \sim \mathcal{GP}(m(\mathbf{x}), \kappa(\mathbf{x}, \mathbf{x}')). \tag{10}$$

The mean function is often taken to be zero, as the mean of the posterior process is not confined to be zero in spite of this. The covariance is typically specified through a kernel function defined on the input domain, $\kappa(\mathbf{x}, \mathbf{x}')$, where the covariance typically decreases with distance. The covariance matrix cannot be defined arbitrarily, as it must be positive semi-definite. In this work, the squared exponential kernel, also known as the radial basis function (RBF), was used. The squared exponential kernel is guaranteed to construct valid covariance matrices, and imposes strong smoothness assumptions on the functions being modelled [34]:

$$\kappa_{\text{SE}}(\mathbf{x}, \mathbf{x}') = \sigma^2 \exp\left(-\frac{\|\mathbf{x} - \mathbf{x}'\|^2}{2l^2}\right). \tag{11}$$

Here, the length-scale $l$ and variance $\sigma$ are hyper-parameters that must either be specified based on domain knowledge or learned from the observed data. The latent function was assumed to be observed with some additional uncertainty, specified through a likelihood function. The likelihood function was chosen to be a multivariate Gaussian distribution described by a diagonal covariance matrix, where each entry on the diagonal corresponds to a wavelength. The variances found in Section 2.5 were used and kept fixed. The slight difference in variance at the bottom and top of the plate was considered to be negligible.

The posterior of a Gaussian process with a Gaussian likelihood function can be exactly inferred in closed form through linear algebra. Exact inference for a Gaussian process with $n$ observations has computational complexity on the order of $\mathcal{O}(n^3)$ and memory requirements $\mathcal{O}(n^2)$. Since the input domain, or regressor, for the latent function is two-dimensional, this quickly becomes restrictive without significant sub-sampling. The poor scaling of Gaussian processes with the size of the dataset is a known challenge, with many approximations being proposed [35]. In this work, a stochastic variational Gaussian process (SVGP) is utilised to circumvent this problem. SVGP uses a sparse or low-rank representation, where a number of inducing inputs are used in place of the full set of observations. The placement of these inducing inputs and kernel hyperparameters is determined through variational optimisation by maximising a lower bound to the true log marginal likelihood [36]. This lower bound can be optimised using stochastic variational inference (SVI), where the variational objective is updated based on noisy gradient estimates from sub-samples of the full dataset (i.e., batches of data) [37,38]. This allows for regression without as much regard for the number of data points.

Gaussian process regression is suitable for this application since the change in light intensity across the calibration plate is a smooth function of height and camera viewing angle. In essence, the correlation in the spatial domain is used to minimise the noise. Figure 5 shows the regression model used in this work, sampled on a grid and compared with a subset of the raw measurements. The latent function being modelled by the Gaussian process is vector-valued with one output dimension per measured wavelength. The hyper-parameters were optimised simultaneously and shared across the output domain. Although adjacent wavelengths will naturally exhibit some correlation, this was not used as an additional regressor in the input domain for two reasons. Firstly, the addition of another dimension to the input space makes the placement of the inducing variables more difficult, and may require additional inducing points compared to the two-dimensional case. Secondly, imposing strong

smoothness assumptions on the measured spectra may not be justified, and may smooth out actual spectral peaks.

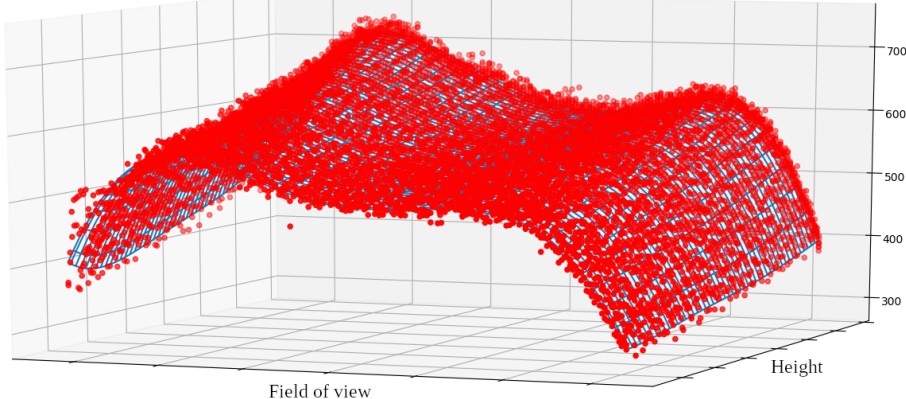

**Figure 5.** Gaussian process reconstruction of digital counts over a tilted reference plate as a function of camera viewing angle ($-26°$ to $26°$) and height above plate (45.4–58 cm) for a single wavelength. The red points are samples, and the blue grid mesh underneath is the reconstruction. A slight difference in the strength of the two lamps is apparent.

## 2.7. Reflectance Calculation

Given a noise-free sample of the latent function $f(x)$, at the height and field of view corresponding to the target material, the albedo or reflectance at a single wavelength can be calculated by dividing $E_{c,m}$ by $E_{c,tr}$ and rearranging, where the subscripts $m$ and $tp$, respectively, denote the unknown material and tilted reference plate.

$$\rho_m^* = \frac{E_{c,m}}{E_{c,\text{tp}}} \frac{\langle \hat{n}_{\text{tp}}, \hat{r}_s \rangle \langle \hat{n}_{\text{tp}}, \hat{r}_c \rangle}{\langle \hat{n}_{\text{m}}, \hat{r}_s \rangle \langle \hat{n}_{\text{m}}, \hat{r}_c \rangle} \rho_{\text{tp}} \tag{12}$$

The measurement of the tilted reference plate, $\rho_{\text{tp}}$, should be made whilst submerged and at the same inclination as used in the tank, to account for any non-Lambertian reflective properties. Alternatively, one can use a flat reference plate in addition to the inclined plate to calculate a second multiplicative correction factor as follows:

$$\rho_m^* = \frac{E_{c,m}}{E_{c,\text{tp}}} \frac{\rho_{\text{fp}}}{\rho_{\text{fp}}^*}. \tag{13}$$

Here, $\rho_{\text{fp}}$ is the measured, or known, reflectance from spectrometer measurements. The term in the denominator, $\rho_{\text{fp}}^*$, is the calculated reflectance of the flat reference plate based on Equation (12), where $E_{c,\text{tp}}$ is sampled from the GP model at the same height as the flat plate. The ratio between the calculated and known reflectance of the flat plate is used to correct for the unknown reflectance of the tilted reference plate, while the tilted plate is used to correct for the changes to attenuation and beam pattern of the lamps as a function of height. The inner products involving the tilted reference plate appear in both expressions and cancel out. The inner product involving the material and flat reference plate cancels out under the assumption that the normal vector of the material is equal to the flat plate. The approach with the flat reference is taken in this work, as the spectrometer measurements are not required to be taken at an angle. If the first approach is taken, a goniometer should be used to measure at the correct angle and distance.

The assumption that the samples exhibit Lambertian responses was not made in this work; we instead acknowledge that the samples were only observed from the nadir direction, and that this is not necessarily representative of the reflectance as seen from sharp angles. No distinction is made

between reflectances captured across the swath of the imager, and the assumption was therefore made that the change in reflectance as a function of incidence angle was negligible within the field of view of the imager. The field of view of the imager was in the range $-26°$ to $26°$, and the samples were centred beneath the camera. Similar concerns have previously been investigated for space-borne hyperspectral imaging, and although not equivalent, it supports this assumption [39].

*2.8. Signal-to-Noise Threshold*

Previously, the noise characteristics were discussed in the context of the regression for the tilted reference plate, where the objective was to verify the assumption of normally distributed noise and subsequently reduce the number of free variables in the optimisation problem. In this section, we briefly discuss the effect of noise when calculating the reflectance for objects of interest. Intuitively, one may think that a higher signal strength of measurements is equivalent to an improvement in the signal-to-noise ratio (SNR). While this implication is true for cases with constant noise properties, it does not necessarily apply when the noise properties vary with the signal level. One such source of noise is *scintillation noise*—a variation in the incident energy on the detector due to mechanical vibrations, changing geometry and/or fluctuation in the medium. This type of noise is proportional to the emitted radiance. Variation in path length and in the reflectance or texture of measured materials may also contribute to the apparent noise. Other sources of noise, such as electrical and thermal noise, may also fluctuate during acquisition. To illustrate how this affects the calculation of the reflectance, we introduce noise terms to Equation (12) with both materials at the same distance and orientation

$$\frac{\rho_m}{\rho_r} + \Delta = \frac{E_{c,m} + \epsilon_m}{E_{c,r} + \epsilon_r}. \tag{14}$$

Here, $E_{c,m}$ and $E_{c,r}$ are measured and reference irradiances with the dark current subtracted, respectively, and the $\epsilon$s are the associated noises for each. If the magnitude of the noise term in the denominator becomes significant in relation to the irradiance $E_{c,r}$, the denominator can fluctuate towards zero, causing the fraction to tend towards infinity. This is liable to occur towards the ends of the spectral range being measured, especially if the light source does not cover the full range of the spectrometer or in the presence of significant attenuation. Obviously, treating these instances as valid reflectance calculations is undesirable and must therefore be detected. This detection should be performed on the individual signals prior to the division. For a comprehensive treatment of noise in spectrometer measurements (see [40]). One source of apparent noise in the measurements is variability in the reflective properties of the target material. By keeping the spectrometer or imager stationary over the target material, one could assume this effect to be constant. This is feasible in a laboratory setting by using a stepping motor instead of a platform moving at constant speed. However, this methodology would not translate well to field applications, where imagers are frequently mounted on moving platforms for spatial coverage.

Calculating the variance directly from the measurements includes the variability of the target material, and overestimates the amount of noise present. Instead, we estimate the variance based on the differences between adjacent wavelengths. Similar approaches have been used previously for spectrometers [41] and magnetic resonance imagery [42]. Taking $X$ and $Y$ to be measurements from two adjacent wavelengths, the variance for the difference of these random variables is

$$\text{Var}\,[X - Y] = \text{Var}\,[X] + \text{Var}\,[Y] - 2\,\text{Cov}\,[X, Y]. \tag{15}$$

The covariance term includes effects which jointly impact the intensity across multiple wavelengths. An estimate of the covariance directly from the measurements is similarly biased due to jointly varying reflectance of the target material. For this reason, the covariance term is replaced

with the correlation (i.e., the covariance scaled by the standard deviation of each variable). This can be calculated over a reference plate, where the reflectance is known to be constant:

$$\text{Corr}\left[X, Y\right] = \frac{\text{Cov}\left[X, Y\right]}{\sigma_X \sigma_Y}. \tag{16}$$

Under the assumption that the standard deviations for adjacent wavelengths $\sigma_X$ and $\sigma_Y$ are approximately equal, inserting Equation (16) into Equation (15) and solving for the average variance of $X$ and $Y$ results in the following expression:

$$\text{Var}\left[X\right] + \text{Var}\left[Y\right] \approx \frac{\text{Var}\left[X - Y\right]}{\left(1 - \text{Corr}\left[X, Y\right]\right)}. \tag{17}$$

The significance of using the correlation instead of the covariance is that the correlation is a scaled measure, and can therefore be used even if the variance of the free variables changes. The correlation can therefore be found once from a target of constant reflectance (e.g., a calibration plate). Note that the variance being estimated contains terms that are both constant and varying with the signal strength, and will therefore not be an unbiased estimate of the variance. This was found to be sufficiently consistent for the purpose of defining a threshold on the signal-to-noise ratio.

In these sections, the experimental setup was outlined. An inclined plate of known reflectance was used to provide a reference intensity for new measurements of unknown materials. A Gaussian process regression model was applied to interpolate these intensity values as a function of height and across-track position while mitigating the noise. Other regression methods could be applied for the same purpose, e.g., smoothing splines. Gaussian process regression has the benefit of having a statistical formulation instead of heuristic smoothing parameters, and was for this reason used. The reflectance was subsequently calculated by dividing new measurements by the reference intensity, corrected for the known reflectance of the reference material. The discussion surrounding noise properties was divided into two sections. The first investigated the assumptions underlying the Gaussian process noise model and obtained fixed values for the variance. The second investigated a threshold on the signal-to-noise ratio, to exclude spectra in areas where the reference measurements or target measurements exhibit low signal strength.

## 3. Results

Rectangular areas seen as distinct lithological endmembers of SMS-related material were selected from the samples. The extracted areas are outlined in Figure 6. While it is possible to select masks based on distance metrics from a reference pixel (e.g., spectral angle mapping (SAM)), we found that defining connected regions gave a better impression of the inherent variability of the materials. All pixels within the marked regions were used for the calculations, except sample C-8, where the most oxidised parts were selected.

Reflectances were calculated using a constant height per sample, measured by hand prior to placing the samples in the tank. The results are displayed in Figures 7 and 8. The spectra from the individual pixels are represented as black dots, and the median spectra in red. The red overlay is where the signal-to-noise ratio was deemed to be too low for accurate reproduction of the true reflectance, with the same threshold being used for all samples.

In terms of the overall reflectance, the samples spanned a range from 3% to 30% and all exhibited an increase in the reflected light towards the upper visible range. Note that the scale varied across the plots, hence the spectra with less variation could appear to be more noisy. Although the general trends of the curves were smooth, there were some interesting features. Area A-3 had a slight depression at about 485 nm and another at 580 nm. The argument can also be made that these features were slightly visible in the same places in area A-2. The sulphides under study (A-2, C-7, C-8, C-9, C-10, and C-11) seemed to generally have a relatively flat response until 490 nm, where they started increasing before

tapering off at 650 nm. The curvature of the heavily oxidised area, C-8, deviated from the others by being slightly concave rather than convex during the ascent. It also did not taper off towards the end of the measured range. The difference between the lower and higher part of the visible spectrum was also greater, producing its orange/red colour to the naked eye. Three measurements of basalt were made, two in a cut portion of the sample, and one at the surface which had been weathered in contact with water. The edge of the cut side of basalt, B-4, had a curve that was close to that of an oxidised sulphide. The basalt did however increase earlier and more steeply, causing a difference even if the spectra were normalised to the same range. The weathered exterior of the basalt, on the other hand, was dark in appearance and was closer to that of an unoxidised sulphide. The basalt was however more linear in its increase than the sulphide, and had a subtle increase towards the blue. The sample was observed to have a slight blue/purple colouration, indicating that this increase was not strictly connected to a low signal-to-noise ratio.

To objectively compare the spectra across different samples, we applied principal component analysis (PCA) and t-distributed stochastic neighbour embedding (t-SNE). PCA finds the orthogonal linear transformation that projects the observations into components of decreasing variance. T-SNE is a nonlinear dimensionality reduction algorithm, especially suited for the visualisation of high-dimensional data on a low-dimensional manifold, while preserving much of the local structure of the high-dimensional data and revealing global structure such as the presence of clusters at several scales [43]. The last property is useful for visualising a wide range of spectra, whereas a linear transformation such as PCA might not reveal structure at all scales. The two-dimensional results are shown in Figure 9, where the spectra are labelled according to their corresponding areas. Note that this information was only used during plotting; the algorithms are completely unsupervised.

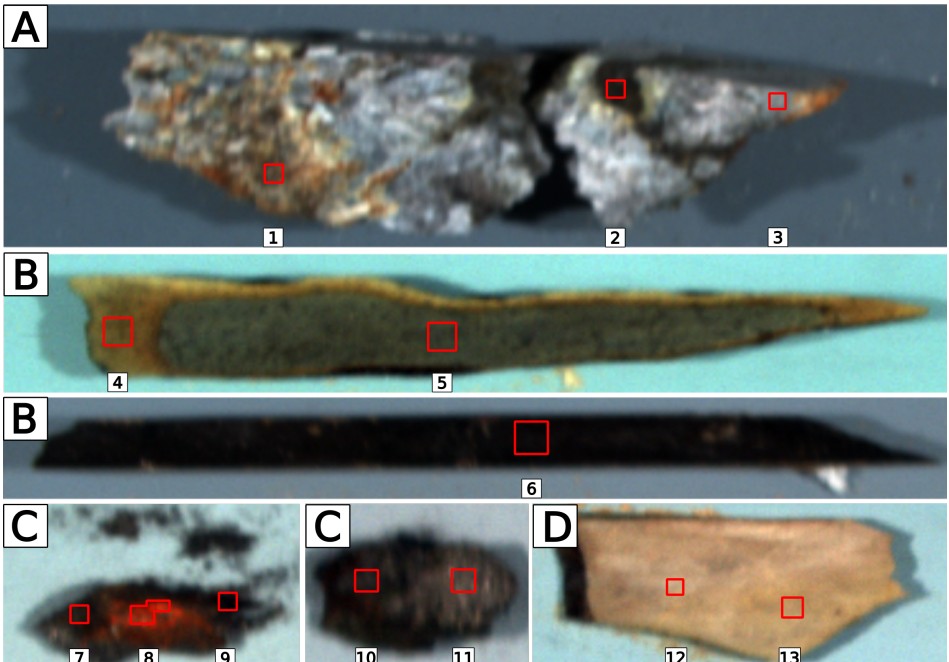

**Figure 6.** Masks for areas used for reflectance curves. The images are RGB representations of the calculated reflectances with channels taken from 667 nm, 540 nm, and 470 nm, respectively. Each channel was normalised per sample to better visualise within-sample differences. The bounding boxes denotes areas in which measurements were averaged to obtain the spectra. The samples are: (**A**) low-grade SMS sample; (**B**) basalt interior and exterior; (**C**) high-grade SMS; and (**D**) mudstone. The dark region between A-1 and A-2 is shadow.

The first two principal components are compared in the top-left plot of Figure 9. The coefficients of the first principal component were almost constant as a function of wavelength; in other words, it primarily captures changes to the average reflectance across the dataset. The coefficients of the second principal component mimic the general trend of the spectra with a smooth increase towards the right. The first and second components capture 95% and 4% of the total variance of the dataset, respectively. The t-SNE algorithm was applied with both squared Euclidean distance and spectral angle distance metrics, and was initialised using PCA. The spectral angle distance metric is defined to be the angle between two vectors both normalised to unit length, $d(\vec{x}, \vec{y}) = \arccos(\langle \vec{x}, \vec{y} \rangle / (\|\vec{x}\|\|\vec{y}\|))$. The Euclidean metric was used both for raw reflectances, and in an instance where the mean distance to the median (i.e., the red points in Figures 7 and 8) was subtracted per area. This removes the variation of the individual spectra which impacts the curve as a whole, compared to the median spectra. The reflectance curves per sample were then more compact, and therefore became more distinct compared to the other samples. The spectral angle metric only looks at the angle between pairs of spectra, and does not take differences in absolute level into account. This represents a loss of information approximately equal to the first principal component. For this reason, it was not as successful in separating the low-dimensional manifold into distinct clusters, but still revealed structure in the dataset. The spectral angle distance metric for t-SNE was included to illustrate the separability if the information in the Euclidean norm is discarded (i.e., the overall intensity). Applying t-SNE with a squared Euclidean distance metric successfully separated the data into different clusters, especially if each sample was centred on the median.

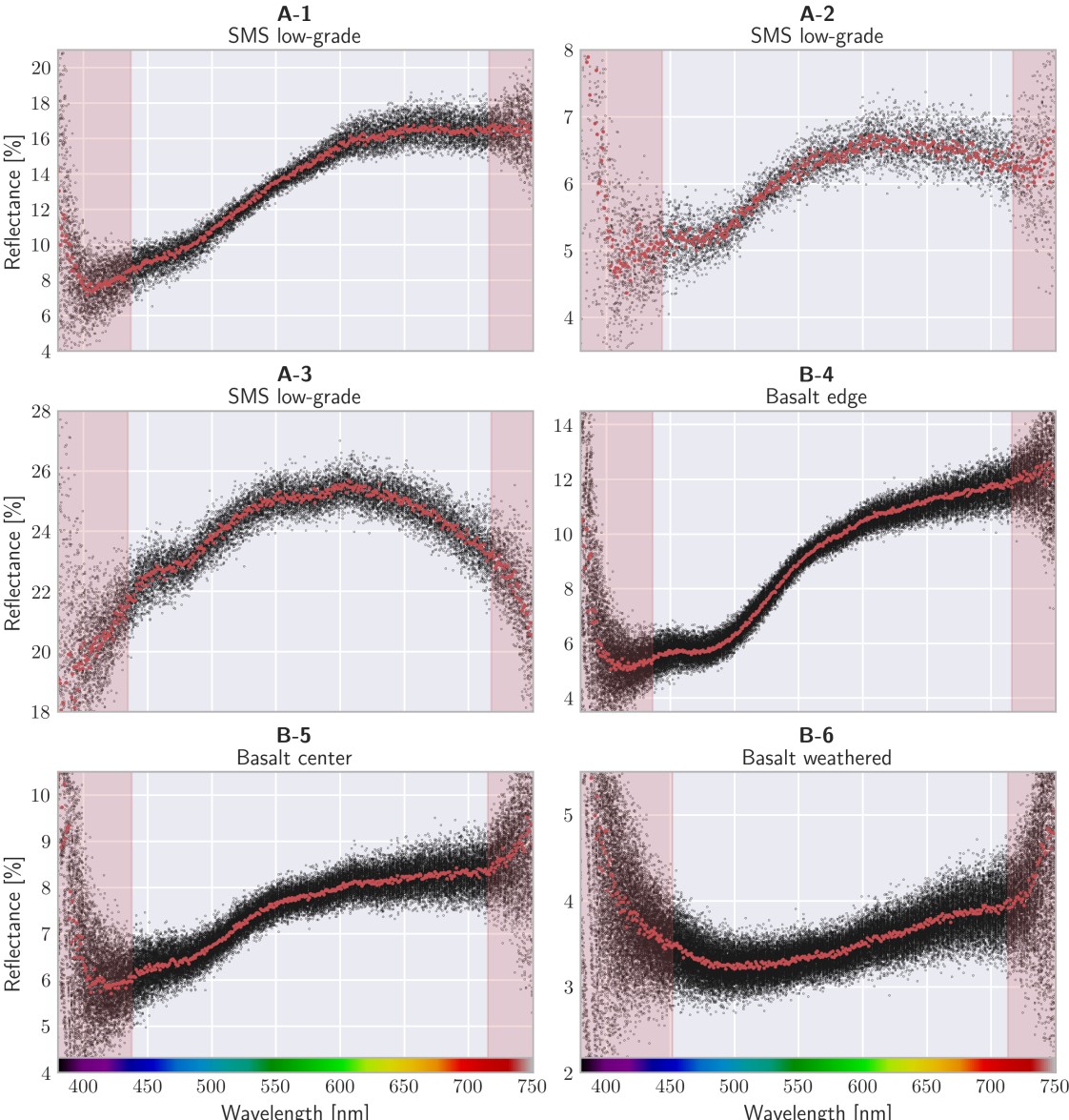

**Figure 7.** Calculated reflectance from the marked areas. The red points are the medians across wavelengths, and the points in black are calculated reflectances for individual pixels centred on the median. The regions in red are regions where the signal-to-noise ratio was below the threshold.

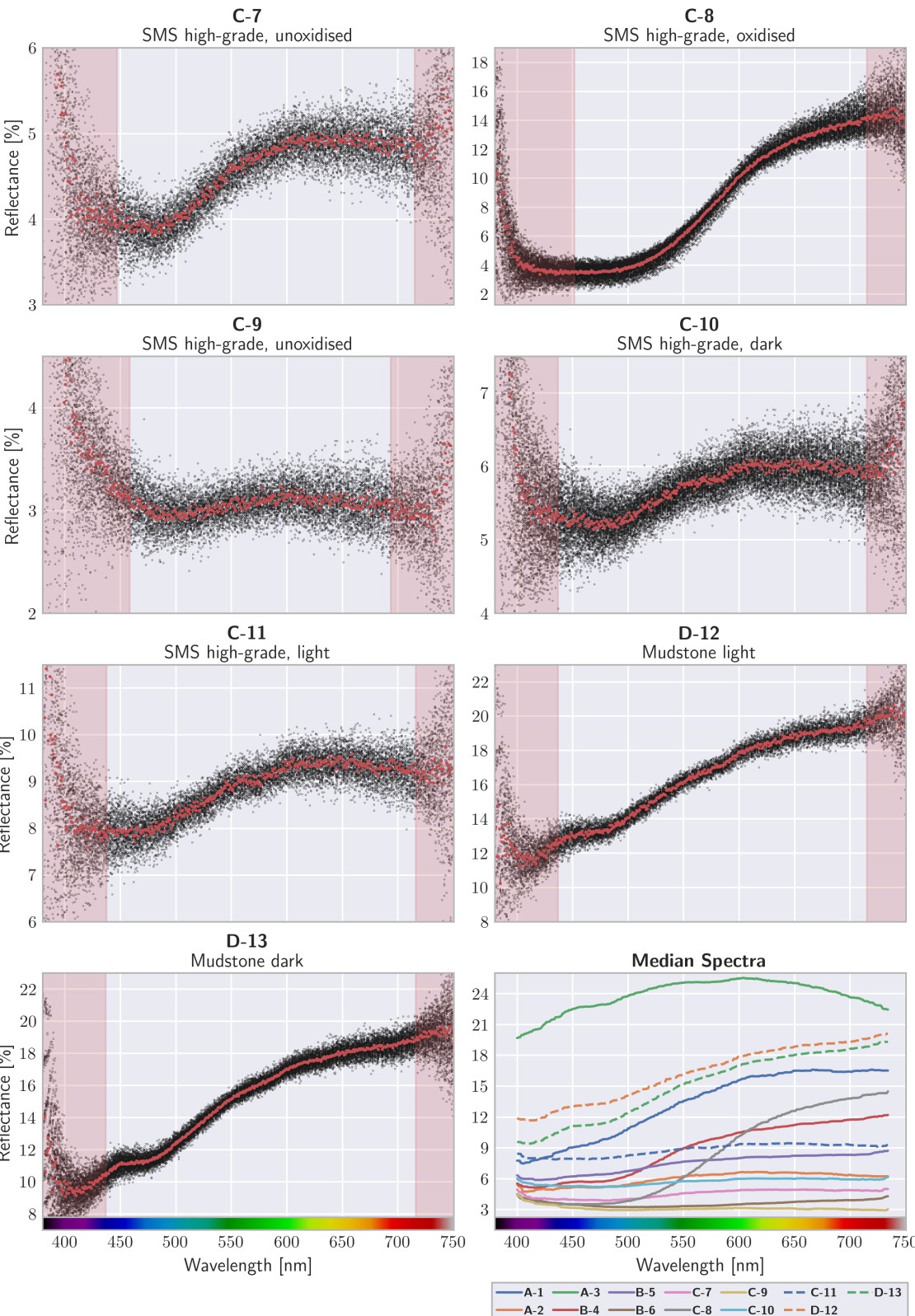

**Figure 8.** Calculated reflectance from the marked areas. The red points are the medians across wavelengths, and the points in black are calculated reflectances for individual pixels centred on the median. The regions in red are regions where the signal-to-noise ratio was below the threshold. In the last plot, a comparison between a selection of median spectra is shown. Note that the x-axes are shared across the plots and the y-axes have individual scales and ranges.

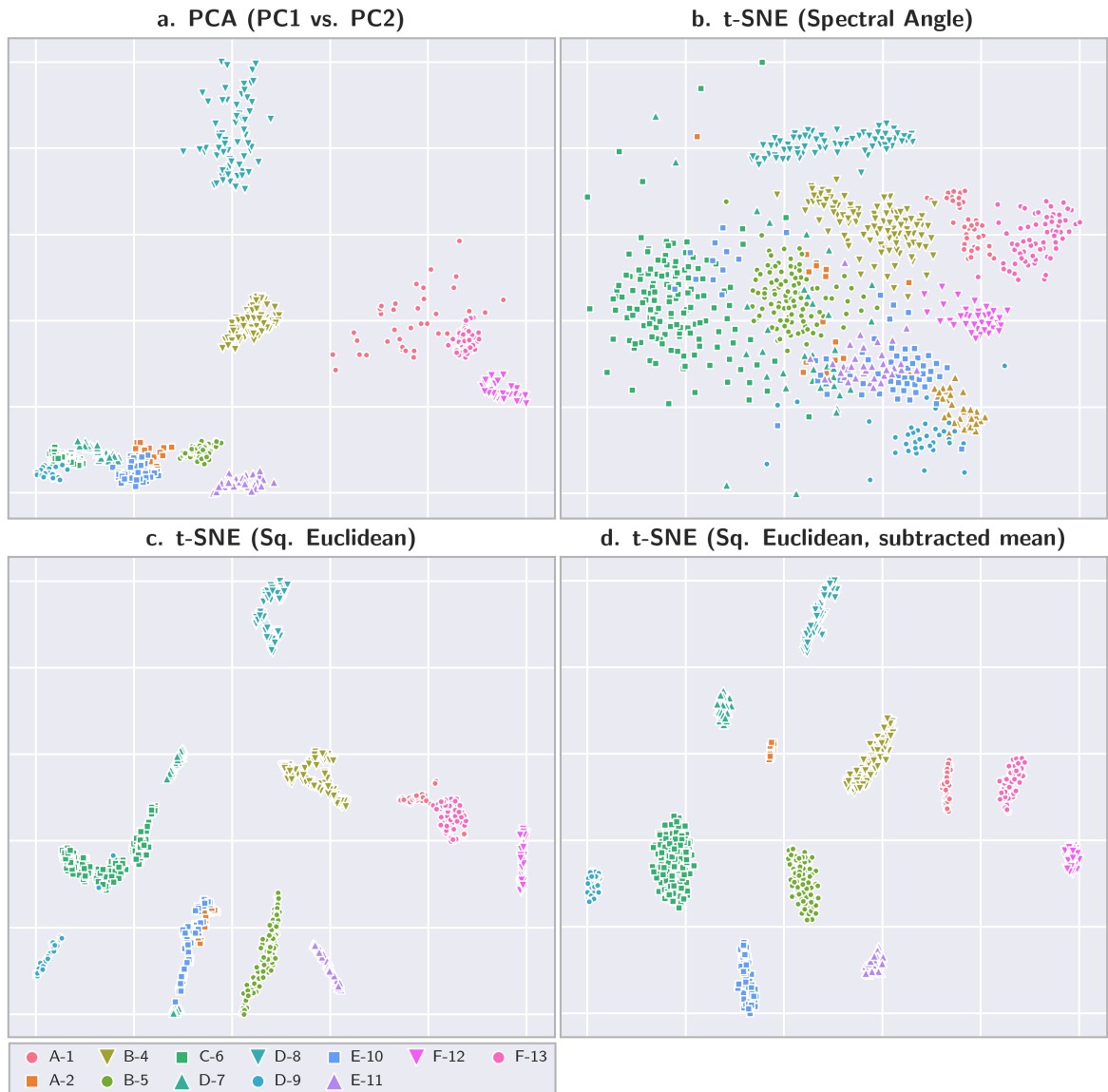

**Figure 9.** Visualisation of all spectra together using dimensionality reduction ($d = 2$). The principal component analysis (PCA) plot has PC1 and PC2 along the *x* and *y* axes, respectively. Visualisation with t-distributed stochastic neighbour embedding (t-SNE) was applied using spectral angle and squared Euclidean pairwise distance metrics. The lower-right plot has the same parameters as the lower left, except each point has the median spectra of that sample subtracted. The points corresponding to sample A-3 are cropped out of the figure, as its distinct spectra causes the plot to be poorly scaled if included.

## 4. Discussion

### 4.1. Experimental Setup

The most important aspect of the setup is the light configuration in relation to the imager. In this setup, two lamps—one on either side of the imager—were used. The first aspect that may be addressed is the choice for the lamps to point straight down. This caused the samples, placed in the middle, to be illuminated by the side of the main axis of the lamp. Given signal strength considerations, one would prefer the sample to coincide with the direction in which the lamp is the strongest. The experimental setup here was intended for imaging at multiple heights, and for this reason pointing the lamps downwards produced a similar beam pattern. For imaging at a fixed height, the lamps could be tilted

inwards, facing the centre of the sample basket. The primary cause of variation with modest changes to sample height was not the attenuation, but the light geometry. Adding diffusers to the lights or using a single-centred lamp in line with the imager could reduce this variation.

In this work, all areas under investigation were directly visible from both light sources. If a sample has significant geometry, areas that are illuminated by a single lamp may appear. Avoiding this situation by sample placement or placement of lights is best, but can also be handled by modelling each light source individually, and considering them to be additive. A line-laser scanner may be added to the setup and run before the hyperspectral acquisition in order to record distance measurements to determine shadow regions.

High wattage halogen lights produce a great deal of heat during operation, and were observed to form bubbles at the glass interface. This is undesirable, and at the very least the bubbles need to escape rather than being trapped on the underside of the lamp. Turning the lamps on/off before and after acquisition may alleviate this problem, but the filaments must reach steady-state operating conditions prior to acquisition.

### 4.2. Field Applicability

The outlined procedure is reasonable for applications in the field, but does require the platform to capture calibration data at different altitudes over a reference plate of known reflectance. This plate does not need to be tilted, since the platform itself can move vertically at known distances, provided that a means of measuring the distance along the camera axis is present. Acoustic instruments or photogrammetry can be applied to obtain distance measurements simultaneously with the hyperspectral acquisition. The methodology presented in this work is still valid for varying distances across the swath and changing the orientation of the platform, if the reference model is formulated in terms of viewing angle and distance.

As always with underwater cameras, providing enough light is key to obtaining good data—preferably in a wide spectral range. While focusing on the blue/green range will allow the platform to maintain a larger distance to the target, important features towards the red part of the spectrum may be lost. A caveat with optical instruments for mineral exploration is the presence of pelagic or re-suspended sediments, which may cover the material of interest. Active vent sites are less afflicted by this due to the ongoing addition of new mineralised material, but chemosynthetic bacterial mats may form instead [44]. Inactive vent sites may be covered in sediments over time, depending on the local sedimentation rates and water currents across the site. In those cases, a platform with intervention capability is needed to remove the covering surface layer (e.g., a suction device).

### 4.3. Spectral Separability

Most of the sample areas in Figure 9 are discriminated into distinct clusters, but their composition is not analysed. Given a larger sample set, one could apply supervised learning to exploit the more subtle shape features for classification purposes and attempt to analyse the samples with respect to different mineralisation. The presence of varying amounts of end-members such as silica/quartz may affect the overall reflectance. If this is the case, one possible outcome is that certain materials can be classified based on prior information such as these reflectance curves, but other materials that are closer in their spectral content may need ground truth materials to train a classifier based on local properties. A calibrated colour camera with a large dynamical range is likely to retain much of the same information as in the first two principal components, but is not able to exploit minute differences in the spectral shape. For reference, the first five principal components had coefficients that seemed to capture some aspect of the spectral shape, whereas the sixth component and above seemed to be tied to noise.

*4.4. Future Work*

Further studies are necessary to expand the available sample body to include other vent sites and end-members. With respect to the presented methodology, a semi-parametric approach may be more suited for field applications, where parts of the model (e.g., the beam pattern) could be kept fixed even if the attenuation coefficients are different. The signal strength remains a challenge, especially for the less-reflective materials such as unoxidised sulphides, and focusing the source lights towards the measurement line is something that can be considered.

## 5. Conclusions

In this paper, we present an experiment performed in a laboratory setting with a hyperspectral imager applied to materials taken from a hydrothermal vent site. A methodology for calculating the reflectances of the materials was proposed along with strategies for noise mitigation. Although the range of wavelengths considered was narrow compared to that used in terrestrial hyperspectral exploration, we observed notable differences between the materials at different stages of oxidisation and across different end-members. The separability of these materials that are typically found at hydrothermal vent sites was found to be distinguishable using unsupervised dimensionality reduction algorithms. The field application of underwater hyperspectral imagery with a set of reference spectra may therefore be a valuable tool for the identification of hydrothermal materials based on surface optical properties.

**Author Contributions:** Conceptualisation, B.S. and Ø.S.; Methodology, Ø.S.; Software, Ø.S.; Validation, Ø.S; Formal analysis, Ø.S.; Investigation, B.S. and Ø.S.; Resources, B.S.; Data curation, Ø.S.; Writing—original draft preparation, Ø.S.; Writing—review and editing, B.S., M.L. and Ø.S.; Visualisation, Ø.S.; Supervision, M.L.; Project administration, B.S. and Ø.S.; and Funding acquisition, M.L.

**Funding:** This research was funded by the Research Council of Norway (Norges Forskningsråd, NFR) grant number 247626 (MarMine).

**Acknowledgments:** We gratefully acknowledge the input and assistance given by Geir Johnsen, Aksel A. Mogstad, and Sigurd A. Sørum during the course of this experiment. The Gaussian processes used in this work were implemented by the open-source projects GPy [45] and GPFlow [46]. The implementation of t-SNE used was from the open-source project scikit-learn [47]. We gratefully acknowledge the input from the anonymous reviewers, which has improved the quality of this article.

**Conflicts of Interest:** The authors declare no conflicts of interest. The funders had no role in the design of the study; in the collection, analyses, or interpretation of data; in the writing of the manuscript; or in the decision to publish the results.

# Appendix A. Whole-Rock Geochemistry

**Table A1.** Table of elements from ICP-ES/MS, reproduced with permission [31].

| Analyte | Au | Mo | Cu | Pb | Zn | Ag | Ni | Co | Mn | Fe | As | U | Th | Sr | Cd | Sb | Bi | V | Ca | P | La | Cr | Mg | Ba | Ti |
|---|---|---|---|---|---|---|---|---|---|---|---|---|---|---|---|---|---|---|---|---|---|---|---|---|---|
| Unit | PPM | PPM | PPM | PPM | PPM | PPM | PPM | PPM | PPM | % | PPM | PPM | PPM | PPM | PPM | PPM | PPM | PPM | % | % | PPM | PPM | % | PPM | % |
| MDL | 0.01 | 0.5 | 0.5 | 0.5 | 5 | 0.5 | 0.5 | 1 | 5 | 0.01 | 5 | 0.5 | 0.5 | 5 | 0.5 | 0.5 | 0.5 | 10 | 0.01 | 0.01 | 0.5 | 1 | 0.01 | 5 | 0.001 |
| Basalt interior Rock Pulp | n/a | 0.5 | 152.1 | 3.8 | 90 | <0.5 | 528.5 | 52 | 2441 | 8.01 | <5 | <0.5 | <0.5 | 95 | <0.5 | <0.5 | <0.5 | 308 | 6.9 | 0.06 | 3.7 | 216 | 4.39 | 68 | 0.91 |
| Basalt edge Rock Pulp | n/a | 1.8 | 261.9 | 6.9 | 119 | <0.5 | 264.3 | 69 | 3271 | 8.03 | 12 | <0.5 | 0.8 | 93 | <0.5 | <0.5 | <0.5 | 319 | 4.65 | 0.06 | 4.9 | 177 | 5.08 | 30 | 0.931 |
| Mudstone Rock Pulp | n/a | 1.5 | 60 | 7.3 | 128 | <0.5 | 281.8 | 19 | 337 | 4.83 | 28 | 2.8 | 9 | 90 | <0.5 | 0.7 | <0.5 | 223 | 0.27 | 0.1 | 31 | 153 | 2.64 | 725 | 0.53 |
| SMS low-grade white Rock Pulp | 5.328 | 3.9 | 56.1 | 122 | 846.8 | 16 | 0.39 | 221 | 0.9 | <0.5 | 630 | <0.5 | 8.3 | <5 | <0.5 | 0.08 | <0.5 | <10 | 76 | 0.02 | 8124 | <1 | 0.02 | <5 | 0.01 |
| SMS low-grade black Rock Pulp | 4.031 | 13.2 | 7647.8 | 13376 | 1818.1 | 341 | 3.6 | 1387 | 2.6 | <0.5 | 321 | 29.5 | 84.2 | 8.3 | <0.5 | 0.05 | <0.5 | <10 | 186 | 0.07 | 252 | <1 | 0.12 | <5 | 0.06 |
| SMS high-grade Rock Pulp | 0.048 | 6.1 | 20457.3 | 37418.1 | 73830 | 18.2 | 469.9 | <1 | 777 | 21.27 | 5 | <0.5 | <0.5 | 5 | 186.4 | 3.1 | 32.1 | <10 | 0.02 | <0.01 | <0.5 | 62 | 0.04 | 91 | <0.001 |

| Analyte | Al | Na | K | W | Zr | Ce | Sn | Y | Nb | Ta | Be | Sc | Li | S | Rb | Hf | Se | Ba | Be | Co | Cs | Ga | Hf | Nb | Rb |
|---|---|---|---|---|---|---|---|---|---|---|---|---|---|---|---|---|---|---|---|---|---|---|---|---|---|
| Unit | % | % | % | PPM | PPM | PPM | PPM | PPM | PPM | PPM | PPM | PPM | PPM | % | PPM | PPM | PPM | PPM | PPM | PPM | PPM | PPM | PPM | PPM | PPM |
| MDL | 0.01 | 0.01 | 0.01 | 0.5 | 0.5 | 5 | 0.5 | 0.5 | 0.5 | 0.5 | 5 | 1 | 0.5 | 0.05 | 0.5 | 0.5 | 5 | 1 | 1 | 0.2 | 0.1 | 0.5 | 0.1 | 0.1 | 0.1 |
| Basalt interior Rock Pulp | 7.62 | 2.27 | 0.04 | 0.6 | 69 | 10 | 1.4 | 32.2 | 2.3 | <0.5 | <5 | 38 | 5.3 | 1.17 | 0.6 | 3.4 | <5 | 73 | 2 | 43.5 | <0.1 | 15.6 | 2.5 | 1.7 | <0.1 |
| Basalt edge Rock Pulp | 6.91 | 2.83 | 0.09 | 1.1 | 56.7 | 14 | 2.3 | 32.7 | 2.5 | <0.5 | <5 | 40 | 17.1 | <0.05 | 1.1 | 2 | <5 | 39 | 5 | 62.6 | <0.1 | 15.1 | 2.7 | 2.1 | 0.8 |
| Mudstone Rock Pulp | 8.58 | 1.45 | 1.97 | 1.7 | 24.3 | 64 | 2 | 9.2 | 15 | 1 | <5 | 15 | 25.3 | 0.14 | 51.9 | 0.5 | <5 | 739 | 2 | 17 | 3.6 | 19.9 | 4.8 | 14.3 | 82.1 |
| SMS low-grade white Rock Pulp | <0.01 | <0.01 | <0.01 | 6.8 | <0.5 | <5 | <0.5 | <0.5 | <0.5 | <0.5 | 0.31 | 0.6 | <0.5 | <0.05 | <0.5 | <0.5 | <5 | <1 | <1 | 0.7 | <0.1 | 0.5 | 6 | 3399.2 | 1.7 |
| SMS low-grade black Rock Pulp | 0.8 | <0.01 | <0.01 | 28.7 | <0.5 | <5 | <0.5 | <0.5 | <0.5 | 0.9 | 5.42 | 2.8 | <0.5 | 91 | 1.8 | <0.5 | <5 | 1.1 | 4.9 | 0.2 | <0.1 | 2.5 | 28 | 874.3 | 0.6 |
| SMS high-grade Rock Pulp | 0.03 | 0.22 | 0.02 | <0.5 | <0.5 | <5 | 10 | <0.5 | <0.5 | <0.5 | <5 | <1 | 12.4 | 18.43 | 1.2 | <0.5 | 633 | 153 | <1 | <0.2 | 0.3 | <0.5 | <0.1 | <0.1 | 0.6 |

| Analyte | Sn | Sr | Ta | Th | U | V | W | Zr | Y | La | Ce | Pr | Nd | Sm | Eu | Gd | Tb | Dy | Ho | Er | Tm | Yb | Lu |
|---|---|---|---|---|---|---|---|---|---|---|---|---|---|---|---|---|---|---|---|---|---|---|---|
| Unit | PPM | PPM | PPM | PPM | PPM | PPM | PPM | PPM | PPM | PPM | PPM | PPM | PPM | PPM | PPM | PPM | PPM | PPM | PPM | PPM | PPM | PPM | PPM |
| MDL | 1 | 0.5 | 0.1 | 0.2 | 0.1 | 8 | 0.5 | 0.1 | 0.1 | 0.1 | 0.1 | 0.02 | 0.3 | 0.05 | 0.02 | 0.05 | 0.01 | 0.05 | 0.02 | 0.03 | 0.01 | 0.05 | 0.01 |
| Basalt interior Rock Pulp | <1 | 102.4 | 0.1 | <0.2 | <0.1 | 313 | 0.7 | 85.1 | 33.9 | 3.4 | 10.3 | 1.73 | 9.9 | 3.32 | 1.4 | 5.02 | 0.95 | 6.07 | 1.36 | 4.1 | 0.59 | 3.71 | 0.56 |
| Basalt edge Rock Pulp | 2 | 101.6 | 0.1 | 0.3 | 0.3 | 316 | 1.5 | 88.2 | 36.1 | 5 | 15.6 | 2.07 | 11.3 | 3.85 | 1.3 | 5.38 | 0.99 | 6.46 | 1.38 | 4.07 | 0.6 | 3.76 | 0.56 |
| Mudstone Rock Pulp | 2 | 102.4 | 1 | 11.6 | 5.1 | 226 | 2.3 | 169.9 | 28 | 43.8 | 90.5 | 9.65 | 35.7 | 6.82 | 1.59 | 6.45 | 0.98 | 5.6 | 1.12 | 3.25 | 0.45 | 3.16 | 0.47 |
| SMS low-grade white Rock Pulp | <1 | 1 | <0.1 | <0.2 | 0.1 | <8 | 0.9 | 0.2 | <0.1 | <0.1 | <0.1 | <0.02 | 1.22 | <0.05 | 0.15 | <0.05 | <0.01 | <0.05 | <0.02 | <0.03 | <0.01 | <0.05 | <0.01 |
| SMS low-grade black Rock Pulp | <1 | 2.8 | <0.1 | 0.6 | 0.2 | <8 | <0.5 | <0.1 | <0.1 | <0.1 | <0.1 | 0.14 | 0.18 | <0.05 | <0.02 | <0.05 | <0.01 | <0.05 | <0.02 | <0.03 | <0.01 | <0.05 | <0.01 |
| SMS high-grade Rock Pulp | 9 | 9.5 | <0.1 | <0.2 | 0.5 | <8 | 1 | 0.1 | <0.1 | 0.2 | <0.1 | <0.02 | <0.3 | <0.05 | <0.02 | <0.05 | <0.01 | <0.05 | <0.02 | <0.03 | <0.01 | <0.05 | <0.01 |

## Appendix B. Whole-Rock Mineralogy

**Table A2.** Mineralogy of samples after XRD analysis, reproduced with permission [31]. In this table + is 0–5%, ++ is 5–5%, +++ is 15% and above. Mineral abbreviations after [48].

| SMS Samples | Description | qtz | po | brt | amo | py | mrc | sp | iso | ccp | gn |
|---|---|---|---|---|---|---|---|---|---|---|---|
| Low grade white | White | | | +++ | + | | | | | | |
| Low grade black | Dark rusty | | + | +++ | ++ | + | + | ++ | + | ++ | + |
| High grade | Dark rusty | +++ | | + | + | + | ++ | +++ | ++ | + | + |

| Non-SMS Samples | Description | qtz | po | ab | chl | aug | gl | ms | mc | | |
|---|---|---|---|---|---|---|---|---|---|---|---|
| Basalt interior, centre | Unaltered | + | | +++ | ++ | +++ | | | | | |
| Basalt interior, edge | Altered | + | | +++ | +++ | +++ | | | | | |
| Mudstone | Massive | +++ | + | ++ | +++ | + | + | ++ | + | | |

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
