# Peer review of "Obtaining Hyperspectral Signatures for Seafloor Massive Sulphide Exploration"

_minerals, doi:10.3390/min9110694_

Round 1
Reviewer 1 Report
This is an interesting paper that introduced an experiment designed to calculate the reflectance of samples. Most of the article focused on the methodology, a brief discussion on the geologic implications might draw more attention from geologists. The methods may need more descriptions so that non-engineers can understand better. The significance of this work may be more justified by the concentration of important metal elements in the SMS, particularly in the samples analyzed, so that the economic value of SMS is highlighted.
I have a question about the experiment setup. The tilted reference plate is used so different height can be achieved under the camera. But how do you measure the height of the scanned sample? Do you have to use a laser scanner/photogrammetry/acoustic to get the height?
I don't quite understand the noise properties discussions. The camera rig was placed stationary over the reference plate, does this mean you took multiple frames of the same line (same height on the tilted plate) and studied the noise between the frames? How many frames were measured? How is the noise normally distributed with "an" identical variance (please add "an" before "identical")? Did you compare the noise in the along-track direction? Can you illustrate this with a figure?
Can you explain what is the spectral angle in t-SNE?
Consider numbering the subplots.
The usage of "however" seems not proper at the end of sentences, in L27, 226, 261, and 306.
L31, is SMS seafloor massive sulphides or submerged massive sulphides?
L32-33 and L33-34, citation?
L43, depths lies?
L56, change into "has also been used"?
Please consider adding an equation after Eq. 5, showing the results of L103-105.
L136-137 mentioned chalcopyrite, isocubanite, sphalerite, pyrite, and marcasite. It would be great if the spectra of these minerals are shown, and possibly identified in the samples.
Fig. 3, how fast is the rig moving above the samples?
Fig. 4, maybe add a subplot to show the ratios of the standard deviations to measured digital counts. Again does this figure show the standard deviations and mean from multiple frames measured on the same line?
L367, blue should be red to be consistent with Figs. 7 and 8.
What are the main differences among the spectra of the marked areas? What geologic/economic meaning can be extracted from these differences? What spectral signatures should we focus on if we use this approach to explore resources? Although this paper claimed that unsupervised methods can distinguish the samples, the meanings of clusters should be presented. There are 13 areas marked and only 9 are shown in the comparison in Fig. 8.
Author Response
Please see the attached pdf for a diff between the versions (note: abstract has some changes not caught by the diff tool).
A brief discussion on the geologic implications might draw more attention from geologists.
The geological implications is that this is an instrument that may be used for identification or analysis based on surface optical properties.This is stated in the conclusion. The instrument may be able to characterise different types of mineralisation, but since the samples under investigation here does not contain different mineralisation, this conclusion cannot be drawn.
Methods may need more descriptions so that non-engineers can understand better
I've tried to add more information pertaining to why the methods are used, to avoid losing readers that are not interested in the details of the methods. I've added a statement that any regression method in fact can be used for this purpose, not strictly Gaussian process regression (although GPs has benefits in terms of their statistical formulation). For the dimensionality reduction algorithm, I would have to add a subsection in the methods part to fully describe them, but I am afraid that this will draw away attention from the core of the paper, as the paper is already quite in-depth technically. PCA is a fairly standard method (although often misused). T-SNE is probably the method that is the most unknown for geologist, but the original author of the method is cited. I've added some additional explanation to why two methods are applied rather than only PCA (t-SNE is non-linear whereas PCA is linear, and t-SNE is therefore able to separate clusters on multiple scales).
How do you measure height of the scanned sample? Do you have to use a laser scanner/photogrammetry/acoustic to get the height?
Mentioned at line 365 (“Reflectances were calculated using a constant height per sample.”)
Changed to the following for clarity “Reflectances were calculated using a constant height per sample, measured by hand prior to placing the samples in the tank.”
The camera rig was placed stationary over the reference plate, does this mean you took multiple frames of the same line (same height on the tilted plate) and studied the noise between the frames?
Correct
How many frames were measured?
50 frames were captured and used for this. Clarified in text
How is the noise normally distributed with "an" identical variance (please add "an" before "identical")?
Agreed. Fixed
Did you compare the noise in the along-track direction? Can you illustrate this with a figure?
The appearent noise in the along-track direction will depend on the material under the line being captured. If the material is homogeneous, such as the tilted plate, this can be investigated. The tilted plate has height variations however, causing variation in the mean intensity. One possible way of doing this would be to subtract the Gaussian process regression of the plate, and compute the variance as a function of along-track position. I didn’t find a good way to visualize this as a function of wavelength, so the compromise I found was to compute the variance at the top and bottom of the plate. If I were to do this again, I would place the imager stationary over the plate in more positions over the plate (e.g. using a stepper motor).
Can you explain what is the spectral angle in t-SNE?
The Euclidean norm is replaced with spectral angle as a distance/similarity metric. The expression used is the following d(x, y) = arccos[ <x, y> / (||x|| ||y||) ]. E.g. arccos of the inner product of the two vectors, both normalized to unit length. The inner product of two unit vectors is equal to the cosine of the scalar angle between them. The definition has been added to the paper.
Consider numbering the subplots.
Done
The usage of "however" seems not proper at the end of sentences, in L27, 226, 261, and 306.
This was caught during the english editing process, but was not uploaded in time for your review. All of these are now fixed.
L31, is SMS seafloor massive sulphides or submerged massive sulphides?
Changed to Seafloor massive sulphides
L32-33 and L33-34, citation?
One possible citation here would be White et al. 2011, “Resource drilling of the solwara 1 seafloor massive sulfide (SMS) deposit”, which details some of the early work of Nautilus minerals. I don’t think this supports the argument enough at this time, seeing as they are facing significant issues with their planned exploitation. The statement has been revised to focus more on what SMS deposits are, rather than whether there is a potential or not.
E.g “Seafloor massive sulphide (SMS) deposits are estimated to contain significant amounts of metal contents \cite{german2016hydrothermal}, and may therefore become important resources in the future.”
The first sentences otherwise are covered by Tivey 2007. I have also added a citation to Shanks and Thurston 2012 “Volcanogenic Massive Sulfide Occurence Model”, which also details some of the same.
L43, depths lies?
Changed to “Such deposits are known to exist at depths between 800 and 5000 m.”
L56, change into "has also been used"?
Fixed
Please consider adding an equation after Eq. 5, showing the results of L103-105.
Done
L136-137 mentioned chalcopyrite, isocubanite, sphalerite, pyrite, and marcasite. It would be great if the spectra of these minerals are shown, and possibly identified in the samples.
Completely agree with this. Performing an experiment such as this with a ground truth from e.g. a QEMSCAN system would be optimal for making this connection between sample and measurement. Snook et Al. 2018 contains a general descriptions of samples from the same vent site as the samples being investigated here, but the samples in this work were not investigated to the same level of detail.
E.g “The sulphides occur in two general forms; as coarser aggregates/grains in aggregates/grains in the black groundmass, and as fine‐grained sulphide lenses the black groundmass, and as fine-grained sulphide lenses that are visible in hand that are visible in hand samples. The lenses are typically composed of approximately 75% pyrite/marcasite, 15% sphalerite, and 10% chalcopyrite/isocubanite. Grains are commonly sub-angular and sized 1–50 μm is usually the largest. Sphalerite and the copper phases are variably intergrown with each other, but pyrite/marcasite is usually the largest. Sphalerite and the copper phases are variably intergrown also occur as distinct grains. Outside of the lens zones, sulphides are usually coarser, and occur as distinct grains. Outside of the lens zones, sulphides are usually coarser, and occur as isolated grains and aggregates of sphalerite, isocubanite, chalcopyrite, and pyrite/marcasite.”
Fig. 3, how fast is the rig moving above the samples?
Measured to be 4.3 cm/s. The frame rate of the imager was set at 16Hz, or 60ms exposure time. The distance traveled per exposure is thus 0.258 cm. Added to methods section
Fig. 4, maybe add a subplot to show the ratios of the standard deviations to measured digital counts. Again does this figure show the standard deviations and mean from multiple frames measured on the same line?
Correct, this is the means and standard deviation over a fixed position at the bottom and top of the inclined reference plate. The meaurements used are at the centre of the plate. I have added the requested subplot
L367, blue should be red to be consistent with Figs. 7 and 8.
Nice catch, the median was originally blue, but red came better across on printed paper. Thank you
What are the main differences among the spectra of the marked areas? What geologic/economic meaning can be extracted from these differences? What spectral signatures should we focus on if we use this approach to explore resources?
Since the sulphides here are from a single vent site, and although at different stages of oxidization, their composition is similar. For me, the most interesting part of these spectra is the ability to exclude some materials as gangue, and the response to oxidisation of sulphides. Additionally, the basalt was observed to have a slightly blue/purple tint when imaged, this is reflected in the spectra for this material with a slight increase towards the lower wavelengths. Since the basalt weathered exterior is similar in its absolute intensity (3% - 5%) to that of an unoxidised sulhpide, it is an interesting result.
Although this paper claimed that unsupervised methods can distinguish the samples, the meanings of clusters should be presented.
Clusters originating from similar samples (e.g. sulphides) are placed close together in the dimensionality reduction plots. The compactness of the clusters and distance to other clusters may be interpreted as a measure of the distinctness of the spectra (at least for PCA, which is a linear method).
There are 13 areas marked and only 9 are shown in the comparison in Fig. 8.
Yes, the idea was to better highlight the differences as two of the samples are much more reflective than the rest. I agree that it is better to show all of them rather than a selection, to avoid confusing the reader. Figure replaced with one containing all spectra

Reviewer 2 Report
Very good paper and very interesting technical discussion: Abstract is misleading: Different types of mineralization are not discussed in this paper Chapter 2: The paper is very technical and the setup is described in detail and extensive, very clear and transparent. Regarding the necessity and connecting to chapter 3 a real link is missing. Not straight forward discussed is the importance and and how chapter 2.1. up to 2.7. are in cooperated in the calculation of the reflectance chapter 2.8. and 2.9 SNR where are the consequences for the measurement and the differentiation of the samples. A concluding para in chapter 2 is missing. Chapter 3: results are visualized in a clear structure. The results are not an analysis of spectral signatures, the results for the different samples describe that the samples can be discriminated, but not analyzed. The description that higher and lower amounts of sulphides can be discriminated which is a very good result. Therefore it should be clearly stated that an discrimination of different mineralization is not performed with this work, as mentioned finally in the conclusions. Text in abstract a conclusion is not consistent. The separability of the measured samples is discussed but no different mineralization. Summary: I want to concentrate on the spectral aspects/ interpretation described in the results: A part of the methodology for the spectral discussion (results) is missing, why did the authors apply different classification algorithms to differentiate the lithology measured , pro-cons? In chapter 2 the setting , data pre processing and correction are comprehensive discussed, but in chapter 3 the methodlogy for the interpretation of the results is only applied. Technology, setup of the measurements are the main aspect of this paper and this should be more clear in the abstract and the title. Methodology in chapter 3 and link to chapter 2 needs to be discussed, here are the supplements not enough.

Author Response
Please see the attached pdf for a diff between the versions (note: abstract has some changes not caught by the diff tool).
Abstract is misleading: Different types of mineralization are not discussed in this paper
Changed to different types of materials (e.g. end-members)
Chapter 2
Regarding the necessity and connecting to chapter 3 a real link is missing. Not straight forward discussed is the importance and and how chapter 2.1. up to 2.7. are in cooperated in the calculation of the reflectance chapter 2.8. and 2.9 SNR where are the consequences for the measurement and the differentiation of the samples.
Right, so 2.1, 2.2, 2.3 details the experimental setup. Added a introductory sentence/paragraph at the start of 2.4 to motivate the need for the inclined plate. Section 2.5 investigates the noise properties of the measurements, with the aim of justifying the assumptions underlying the Gaussian process regression model (Gaussian likelihood/noise model with variable variance across wavelengths). Added a paragraph at the start of 2.6 to motivate the need for a regression model (interpolation + noise mitigation). Also added a concluding paragraph after ch2, which also outlines that other regression models are usable for this purpose (e.g. splines), but that Gaussian process is well suited for this purpose, as one can work in a statistical sense, instead of tuning smoothness parameters of splines manually.
A concluding para in chapter 2 is missing.
Added. Hopefully this concluding paragraph ties into chapter 3.
Chapter 3
Results for the different samples describe that the samples can be discriminated, but not analyze
I've tried to clarify this more in the discussion
It should be clearly stated that an discrimination of different mineralization is not performed with this work, as mentioned finally in the conclusions
Added a sentence regarding this under Discussion – Spectral Separability. Mentioned as a possible future application, but a conclusion cannot be drawn here wrt. this
Conclusion
Text in abstract a conclusion is not consistent. The separability of the measured samples is discussed but no different mineralization. Agreed, changed abstract to account for this
Summary
I want to concentrate on the spectral aspects/ interpretation described in the results
A part of the methodology for the spectral discussion (results) is missing, why did the authors apply different classification algorithms to differentiate the lithology measured , pro-cons?
Added some more information regarding these methods in the results section. If desired, these can be separated out in a section in the methods section. In essence the two different dimensionality reduction algorithms differ in that PCA is a linear method placing emphasis on the variance of the dataset, whereas t-SNE is a non-linear method which attempts to minimize the distance between the join probability of the low-dimensional embedding and high-dimensional data using the Kullbeck-Leibler divergence (iteratively). One benefit is that clusters at different scales are more easily visualized, but the distances between the clusters cannot be compared directly (due to the non-linear property). The spectral angle distance metric is used to illustrate what happens if information pertaining to the absolute intensity of the spectra is discarded (essentially discarding approximately 95% of the information in PCA). The final plot (d) illustrates the separability if the mean variation about the median is removed (i.e. the variation in each cluster is only due to shape differences per area, not variation in mean intensity). The final plot may be removed if so desired, as it doesn’t contribute to the overall discussion.
In chapter 2 the setting , data pre processing and correction are comprehensive discussed, but in chapter 3 the methodlogy for the interpretation of the results is only applied.
Yes, the dimensionality reduction was included to discuss the separability of the samples, but the methodology is simply cited from it’s original paper. I have added a reference to the implementation used for t-SNE in acknowledgements.
Technology, setup of the measurements are the main aspect of this paper and this should be more clear in the abstract and the title.
Changed title, and altered abstract slightly to place emphasis on the methods
Methodology in chapter 3 and link to chapter 2 needs to be discussed, here are the supplements not enough.
Is the paragraph at the end of ch2 sufficient to link chapter 2 and 3?
